

# Quantifying the contribution of anthropogenic influence to the
# East Asian winter monsoon in 1960–2012
Xin Hao*[1,2], Shengping He[4,1], Huijun Wang[1,2,3], Tingting Han[1,2]
*[1] Collaborative Innovation Center on Forecast and Evaluation of Meteorological*
*Disasters/Key Laboratory of Meteorological Disaster, Ministry of Education, Nanjing*
*University for Information Science and Technology, Nanjing 210044, China*
*[2] Nansen-Zhu International Research Centre, Institute of Atmospheric Physics, Chinese*
*Academy of Sciences, Beijing 100029, China*
*[3] Climate Change Research Center, Chinese Academy of Sciences, Beijing100029, China*
*[4] Geophysical Institute, University of Bergen and Bjerknes Centre for Climate Research,*
*Bergen 0025, Norway*
*Corresponding author: Xin Hao, haoxlike91@163.com*
Abstract
The East Asian winter monsoon (EAWM) can be greatly influenced by many factors that can
be classified as anthropogenic forcing and natural forcing. Here we explore the contribution of
anthropogenic influence to the change in the EAWM over the past decades. Under all forcings
observed during 1960–2013 (All-Hist run), the atmospheric general circulation model is able to
reproduce the climatology and variability of the EAWM-related surface air temperature and 500
hPa geopotential height, and shows a statistically significant decreasing EAWM intensity with a
trend coefficient of ~ −0.04 yr$^{-1}$ which is close to the observed trend. By contrast, the simulation,
which is driven by the same forcing as All-Hist run but with the anthropogenic contribution to
them removed, shows no decreasing trend in the EAWM intensity. By comparing the simulations
under two different forcing scenarios, we further reveal that the responses of the EAWM to the
anthropogenic forcing include a rise of 0.6℃ in surface air temperature over the East Asia as well
as weakening of the East Asia trough, which may result from the poleward expansion and
intensification of the East Asian jet forced by the change of temperature gradient in the



troposphere. Additionally, compared with the simulation without anthropogenic forcing, the
frequency of strong (weak) EAWM occurrence is reduced (increased) by 45% (from 0 to 10/7).
These results indicate that the weakening of the EAWM during 1960–2013 may be mainly
attributed to the anthropogenic influence.
**Key words**: anthropogenic influence, East Asian winter monsoon, contribution

## 1.  Introduction

The East Asian winter monsoon (EAWM) is one of the most dominant climate

systems in East Asia. It greatly affects the disastrous winter weather such as cold
waves, snowstorms, air pollutions, and spring duststorms (Li et al., 2016; Li and
Wang, 2013; Wang et al., 2009; Zhou et al., 2009; Chang et al., 2006). Prominent
circulation components from surface to the upper troposphere associated with
temperature condition during the boreal winter are dynamically linked to the EAWM.
At surface, the EAWM contains the cold Siberian high dominated over the East Asian
continent and the warm Aleutian low located in the high-latitude North Pacific, which
accompanies with prevailing northwesterly winds in the low-level troposphere (He
and Wang, 2013; Wang and Jiang, 2004; Gong et al., 2001; Guo 1994; Lau and Li,
1984). At 500 hPa locates the East Asian trough which determines the outbreak and
intensity of the EAWM (He et al., 2013; Cui and Sun, 1999; Sun and Li, 1997). In the
upper troposphere, a key component of the EAWM is the East Asian jet with its
maximum core being located to the southeast of Japan (Jhun and Lee, 2004; Boyle
and Chen, 1987). Concurrent with the change of these atmospheric circulation, the
change of winter surface air temperature (SAT) over East Asia is closely related to the
variation in the EAWM (Hao and He, 2017; Lee et al., 2013; Wang et al., 2010).

Above primary components of the EAWM system are subject to obvious changes

under the influence of global warming (e.g., Li et al., 2018; Li et al., 2015; IPCC,
2013; Hori and Ueda, 2006; Kimoto, 2005; Zhang et al., 1997). Under different global
warming scenarios, thermodynamic contrast between the East Asian continent and the
Pacific Ocean is reduced uniformly characterized with weakening of the East Asian
trough (EAT) as well as the East Asian jet, indicating a weakening of the EAWM (e.g.,





Xu et al., 2016; Kimoto, 2005). Previous studies based on Coupled models generally
agree on the effect of global warming on the EAWM (Hong et al., 2017; Xu et al.,
2016; Kimoto, 2005; Hu et al., 2000). However, previous studies mainly conduct
qualitative research on the potential influence of the global warming, it's still unclear
to what extent can the anthropogenic activities impact the EAWM. This study aims to
quantitatively estimate the contribution of increasing anthropogenic emissions over
the past decades to the change of the EAWM, which is essential for the projection of
the EAWM in the future.

2.  Data and Method

Monthly mean dataset including SAT, 500 hPa geopotential height and 250 hPa

zonal wind is obtained from National Center for Environmental Prediction/National
Center for Atmospheric Research (NCEP/NCAR) Reanalysis 1 dataset at a horizontal
resolution of $2.5° \times 2.5°$ (Kalnay et al., 1996). Hereafter it is referred to as
"observations". To explore the contribution of the anthropogenic emissions to climate
change, two different simulations from the C20C+ Detection and Attribution Project
(http://portal.nersc.gov/c20c/data.html) are compared in the context of two different
forcing scenarios. One is the **All-Hist** which was forced with time-vary boundary
conditions (e.g., greenhouse gas concentrations, anthropogenic and natural aerosols,
ozone, solar luminosity, land cover, sea surface temperatures and sea ice) observed
during the past few decades. The other is the **Nat-Hist** which was forced with
observed sea surface temperature and sea ice concentrations from which the
anthropogenic    contribution    has    been    removed    (please    refer    to
http://portal.nersc.gov/c20c/data.html for more details). Meanwhile, the natural
external forcing such as greenhouse gas concentrations and aerosols was set to
preindustrial levels. We analyses the simulations by an atmospheric general
circulation model HadGEM3-A-N216 (Christidis et al., 2013; approximately $0.56° \times$
$0.83°$ horizontally) available from the C20C+ Detection and Attribution, which has
been used to conduct the above two sets of experiments from 1960 to 2013. Both



All-Hist and Nat-Hist runs include 15 ensemble members. Each realization in the two scenarios differs from the other only in its initial state. The ensemble-mean of the runs number 1, 2, 5, 13, 14, and 15 (which show a better performance in simulating interannual, decadal and linear trend change of EAWM) under the All-Hist scenarios agrees best with the reanalysis dataset (such as climatology, interannual and decadal change of EAWM; evaluation of other runs of model shown in supplementary). Therefore, the simulations of these 6-members ensemble are used in this study.

In this study, we focus on the winter mean which is the average of December, January and February (e.g., the winter 2008 refers to the boreal winter of 2008/2009). Two intensity indices are used to describe the variability of the EAWM: one is defined as the area-averaged height geopotential at 500 hPa in 35 °–45 °N, 125 °–145 °E (EAWMI_HGT; Sun and Li, 1997); the other is defined as the area-averaged SAT in 25 °–45 °N, 105 °–145 °E (EAWMI_SAT; Lee et al., 2013). Both area-averaged values are multiplied by -1 so that positive values correspond to strong EAWM; 9-year running mean of the index represents the interdecadal variability of the EAWM.

## 3. Results and Discussions

### 3.1 Assessment of the atmospheric circulation pattern simulated by model in All-Hist runs

The EAWM is characterized by northerly winds over East Asia, the Siberian high, the Aleutian low, the deep East Asian trough, the upper tropospheric East Asian jet stream, as well as the cold and dry conditions over East Asia (e.g., Hao et al., 2016; Lee et al., 2013; He and Wang, 2013; Wang and Jiang, 2004; Sun and Li, 1997). In this study, the performance of the HadGEM3-A-N216 model in simulating the above characteristics of the EAWM is firstly evaluated by comparing the corresponding results in the All-Hist runs with reanalysis dataset in the period of 1960–2012.

Figures 1a-d show the climatology of the SAT and 500 hPa geopotential height in winter from the observations and simulations in the All-Hist run. The winter SAT climatology over East Asia in simulations (Fig. 1a) is generally consistent with the



observed counterpart (Fig. 1b). The model has successfully reproduced the dominant
features of East Asian winter SAT such as the northwest-to-southeast temperature
gradient, the 0℃ isotherm of SAT stretching from western China (around 27.5 °N)
northeastward to north Japan (around 42.5 °N), the cold center located over the
Tibetan Plateau (Figs. 1a and 1b). Compared with the observations, the simulated SAT
shows apparent cold bias over the north of 40 °N but less bias over the south of 40 °N.
In the middle troposphere, the main features (position of axis and intensity) of the
EAT are also generally reproduced by the model. The simulated SAT in 25 °–45 °N,
105 °–145 °E (Lee et al., 2013) and 500 hPa geopotential height in 35 °–45 °N, 125 °–
145 °E (Sun and Li, 1997) used for the EAWM indices show high spatial correlations
with the observations (Fig. 1e), which are exceed 0.99. Additionally, high spatial
correlations of the simulated SAT and 500 hPa geopotential height with the
observation are accompanied by small root mean square errors (Fig. 1e). It means that
the All-Hist runs have well simulated the EAWM climatology.

131   The variability of the EAWM is also compared between the simulations and the

observations. It is found that the correlations between the simulated EAWM indices
and the observed EAWM indices are 0.3 for EAWMI_SAT and 0.31 for
EAWMI_HGT, respectively (Fig. 2), which are statistically significant. Additionally,
the interdecadal variability of the EAWM indices are closely correlated between the
simulations and the observation with correlation coefficients of 0.7 for EAWMI_SAT
and 0.76 for EAWMI_HGT (Fig. 2). The result suggests that the All-Hist runs have
well simulated the interannual and interdecadal variability of the EAWM and can be
further used to investigate the anthropogenic impact on the EAWM.

3.2 Contribution of anthropogenic influence to the East Asian winter monsoon

142   To investigate the anthropogenic contribution to the change of the EAWM, we

compare the EAWM in the All-Hist runs with those in the Nat-Hist runs. Both of the
EAWM indices in the All-Hist runs show statistically significant decreases over the
past decades, with trend coefficients of $-0.044$ $(yr^{-1})$ and $-0.038$ $(yr^{-1})$, respectively,
which are similar to the observed trends (-0.023 and -0.02, respectively; Fig. 2). By



contrast, the EAWM indices in Nat-Hist run show an increasing trend, instead (Fig. 2).
It suggests that the increasing anthropogenic emissions in the past decades may
contribute to the weakening of the EAWM.
Figure 3 displays the composited differences of the simulated winter SAT and
500 hPa geopotential height between the All-Hist runs and in the Nat-Hist runs, which
approximately reflect the response of the EAWM to anthropogenic forcing. The
composited differences show clearly that winters with anthropogenic forcing see
apparent warmer anomalies over most parts of East Asia except for southeast China as
well as warmer conditions over the western North Pacific (Fig. 3a). Such a response is
similar to the one revealed by previous CMIP5 studies (Hong et al., 2017; Xu et al.
2016). Xu et al. (2016) suggested that the large positive anomalies over high-latitude
western North Pacific are due to a north ward shift of the significantly intensified
Aleutian low induced by the melting sea ice in the Bering Sea and Okhotsk Sea (Gan
et al., 2017). Quantitatively, compared with the situation without anthropogenic
influence, the wintertime SAT averaged over ($20°$–$60°N$, $100°$–$140°E$) increases by
$0.6°C$ over the last half-century due to anthropogenic influence (Fig. 3a). At middle
troposphere, responses of the 500 hPa geopotential height to anthropogenic forcing
shows obviously positive anomalies over East Asia with a value of 15.7 m, implying a
shallower EAT which results in less powerful cold air to East Asia (Fig. 3b). The
model simulations indicate clearly that the anthropogenic influence may induce a
weaker EAWM.
It should be noted that, in the low-level troposphere, the high-latitude warming
induced by the anthropogenic forcing is apparently stronger than the warming at
lower-latitudes (Fig. 4a), which is the so-called "polar amplification" (Meehl et al.,
2007; Collins et al., 2013). Meanwhile, in the high-level troposphere, obviously
warming occurs over the tropical regions and the Arctic, but cooling occurs over the
high-latitude regions under the anthropogenic influence (Fig. 4a). As a result, a
broadening and intensifying Hadley circulation appears, which is consistent with the
observed phenomena revealed by previous studies that a poleward expansion and
intensification of the winter Hadley circulation in the past few decades (Hu and Fu,




2007; Mitas and Clement, 2005; Hu et al., 2005). Such a change in the Hadley
circulation implies a poleward shift of the East Asian jet (Fig. 4b), together with a
reinforcement and expansion of Western Pacific subtropical high and an increase of
SLP in the high-latitude East Asia (Fig. 4c). The change of SLP also indicates a weak
decrease of the Siberian high and an intensified Aleutian low. Thus, under the
anthropogenic influence, significant easterly anomalies occur in the mid- and
high-latitude of East Asia and significant southerly anomalies occur in the
low-latitude of East Asia (Fig. 4c), leading to a subdued EAWM. We further explore
the contribution of anthropogenic influence to the occurrence of strong/weak EAWM.
The case with the normalized index larger than 1.0 (smaller than −1.0) is defined as a
strong (weak) EAWM event. The number of the strong/weak EAWM events is shown
in Fig. 5. The two observed EAWM indices display 10 (8) and 9 (9) strong (weak)
EAWM events during 1960–2012, respectively. Interestingly, the two simulated
EAWM indices in the All-Hist run display 11 (10) and 11 (7) strong (weak) EAWM
events, respectively. The number of strong or weak EAWM events forced by the
observed time-varying boundary conditions during the past few decades (All-Hist run)
is very close to the number in observations. However, during 1960–2012, the
simulated two EAWM indices in the Nat-Hist runs display 21 (0) and 19 (0) strong
(weak) EAWM events, which is remarkably different from the number in the All-Hist
runs as well as the observations. It implies that, in the past decades, the frequency of
occurrence of strong EAWM may have reduced by 45% due to the anthropogenic
forcing and the anthropogenic forcing is a dominant contributor to the occurrence of
weak EAWM.

## 4   Conclusion

The contribution of the anthropogenic influence to the climatology, trends, and
the frequency of occurrence of strong/weak EAWM is explored in this study based on
numerical simulations. Firstly, we evaluate the performance of the climate model
(HadGEM3-A-N216) in simulating the climatology of wintertime circulation over



East Asia and variation of EAWM indices during 1960–2012. The winter-mean states
of SAT and 500 hPa geopotential height related to the EAWM in the All-Hist runs
resemble well those in observation with spatial correlation coefficients of greater than
0.99. Also, the interannual and interdecadal variation of the EAWMI_HGT and
EAWMI_SAT can be well reproduced by the model under All-Hist scenario. Because
of the well performance of the All-His runs in simulating the EAWM indices and
winter-mean atmospheric circulation over the East Asia, the exploration about
changes of the EAWM induced by anthropogenic influence is considered reliable.

Under All-Hist scenario, the EAWM indices have significantly decline trends

over the past decades, which are consistent with those in observations, indicating that
the weakening of the EAWM could be simulated by the climate model with all forcing.
However, the EAWM indices do not have such trends in the Nat-Hist runs. Compared
the area-averaged SAT and 500 hPa geopotential height related to the EAWM for the
period of 1960–2012 between two families of experiments, it is found that
anthropogenic emissions induce obviously positive SAT anomalies in the most region
of East Asia and a weakened EAT, as shown in previous results (Hu et al., 2000; Hori
and Ueda, 2006; Xu et al. 2016; Hong et al., 2017; Hong et al., 2017). Additionally,
11 (11) strong EAWM events and 10 (7) weak EAWM events are forced by All-Hist
scenario during 1960–2012, which are close to the frequency of occurrence of strong
and weak EAWM in observations, while 21 (19) strong EAWM events and 0 (0) weak
EAWM event are forced by Nat-Hist scenario. Overall, under anthropogenic influence,
during 1960–2012, the EAWM continued to be weakened, and the frequency of
occurrence of strong (weak) EAWM had decreased (increased) by 45% (from 0 to
10/7). The poleward expansion and intensification of East Asian jet induced by
anthropogenic influence may be the reason for the weakening of the EAWM. A
decrease trend is found both in observation and in the All-Hist runs, therefore more
attention should be given to the EAWM variability under anthropogenic influence.

**Author contributions.** Xin Hao conceived the idea for the study and wrote the paper.
All authors contributed to the development of the method and to the data analysis.



**Acknowledgements**

This work was supported by the National Science Foundation of China (Grant

41421004, 41875118, 41605059 and 41505073). All datasets can be accessed publicly.
The NCEP analysis dataset can be downloaded from
https://www.esrl.noaa.gov/psd/data, and the simulations can be downloaded from
http://portal.nersc.gov/c20c/data.html.

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

Figure caption:

Figure 1 Climatology of winter-mean (DJF) (a) surface air temperature (shading, ℃)

(c) 500 hPa geopotential height (shading, m) during 1960–2012, based on NCEP

reanalysis data. (b), (d) As in (a), (b), but for the model's All-Hist runs. (e)

Taylor diagram of winter-mean climatology for surface air temperature (TAS;

25 °–45 °N, 105 °–145 °E ) and 500 hPa geopotential height (H500; 25 °–45 °N,

105 °–145 °E). The rectangle marks the areas used to calculate the climatology in

taylor diagram.

Figure 2 (a) The time series of the normalized EAWMI_SAT (curve) and their linear

trend (line) during 1960–2012, based on NCEP reanalysis dataset (top), outputs

of model in All-Hist run (middle), and outputs of models in Nat-Hist run

(bottom). (b) As in (a), but for the EAWMI_HGT. "tr" is an abbreviation for

"linear trend coefficient". "*"means the tr is significant at 95% confidence level

based on the Mann-Kendall test, and "'" means the tr is significant at 90%

confidence level. "cor" is an abbreviation for "correlation coefficient between

simulated EAWM index under All-Hist scenario and observed EAWM index",

"cor_dec" is an abbreviation for "correlation coefficient in decadal time-scale".

Note that the time series of the EAWM indices base on outputs of model in the

Nat-Hist runs are standardized by the climatology simulated by the All-Hist runs.



Figure 3 Composite differences of winter-mean (a) surface air temperature

(shading, ℃) and (b) 500 hPa geopotential height (shading, m) between the

All-Hist runs and Nat-Hist runs, during 1960–2012. The plus signs denotes

where the composite differences are significant at the 95% confidence level

based on two-sided Student *t* test.

Figure 4 Composite differences of winter-mean (a) air temperature (shading, ℃) over

90 ℃E–150 ℃E, (b) 250 hPa zonal wind (shading, m/s) and (c) sea level pressure

(shading, hPa) and 850 wind (vector, m/s) between the All-Hist runs and

Nat-Hist runs, during 1960–2012. Red contours denote the climatology of

All-Hist runs. The plus signs denotes where the composite differences are

significant at the 95% confidence level based on two-sided Student *t* test.


Figure 5 (a) The number of strong EAWM events during 1960–2012, based on NCEP

reanalysis dataset (left), outputs of model in the All-Hist runs (middle), and

outputs of model in the Nat-Hist runs (right). (b) As in (a), but for weak EAWM

events.

















Figures

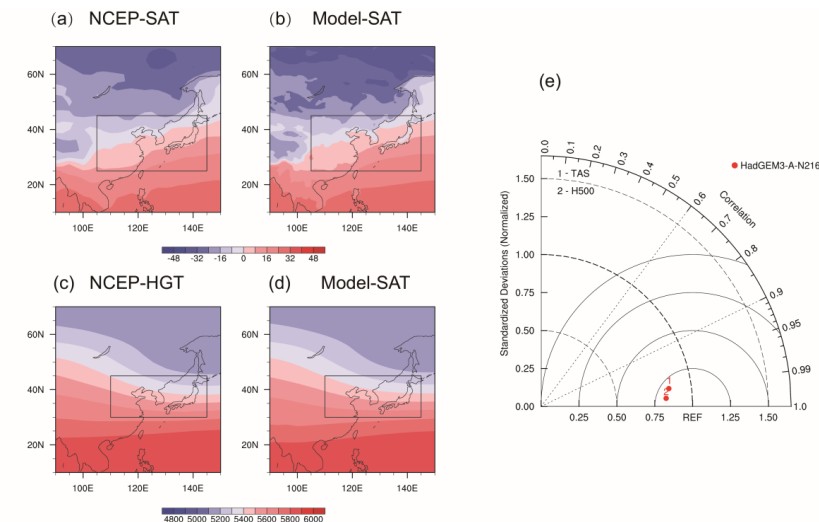

Figure 1 Climatology of winter-mean (DJF) (a) surface air temperature (shading, ℃)

(c) 500 hPa geopotential height (shading, m) during 1960–2012, based on NCEP

reanalysis data. (b), (d) As in (a), (b), but for the model's All-Hist runs. (e)

Taylor diagram of winter-mean climatology for surface air temperature (TAS;

25 °–45 °N, 105 °–145 °E) and 500 hPa geopotential height (H500; 25 °–45 °N,

105 °–145 °E). The rectangle marks the areas used to calculate the climatology in

taylor diagram.

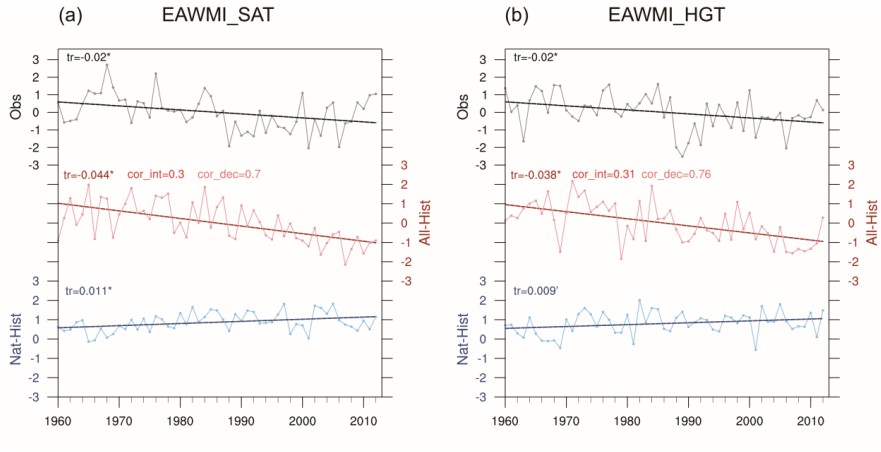

Figure 2 (a) The time series of the normalized EAWMI_SAT (curve) and their linear

trend (line) during 1960–2012, based on NCEP reanalysis dataset (top), outputs





of model in All-Hist run (middle), and outputs of models in Nat-Hist run
(bottom). (b) As in (a), but for the EAWMI_HGT. "tr" is an abbreviation for
"linear trend coefficient". "∗"means the tr is significant at 95% confidence level
based on the Mann-Kendall test, and "′ " means the tr is significant at 90%
confidence level. "cor" is an abbreviation for "correlation coefficient between
simulated EAWM index under All-Hist scenario and observed EAWM index",
"cor_dec" is an abbreviation for "correlation coefficient in decadal time-scale".
Note that the time series of the EAWM indices base on outputs of model in the
Nat-Hist runs are standardized by the climatology simulated by the All-Hist runs.

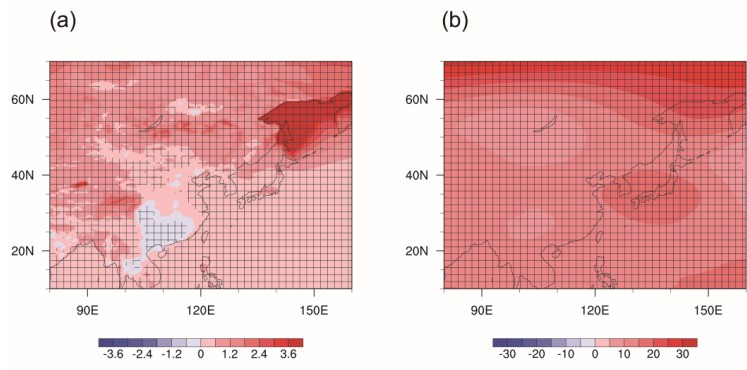


Figure 3 Composite differences of winter-mean (a) surface air temperature
(shading, ℃) and (b) 500 hPa geopotential height (shading, m) between the
All-Hist runs and Nat-Hist runs, during 1960–2012. The plus signs denotes
where the composite differences are significant at the 95% confidence level
based on two-sided Student *t* test.

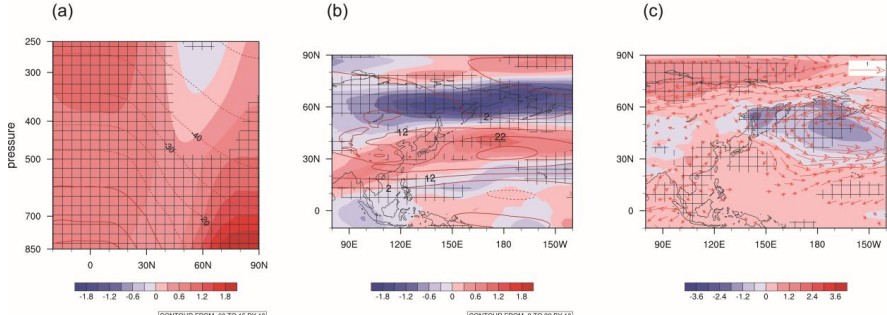


Figure 4 Composite differences of winter-mean (a) air temperature (shading, ℃) over



90°E–150°E, (b) 250 hPa zonal wind (shading, m/s) and (c) sea level pressure
(shading, hPa) and 850 wind (vector, m/s) between the All-Hist runs and
Nat-Hist runs, during 1960–2012. Red contours denote the climatology of
All-Hist runs. The plus signs denotes where the composite differences are
significant at the 95% confidence level based on two-sided Student $t$ test.

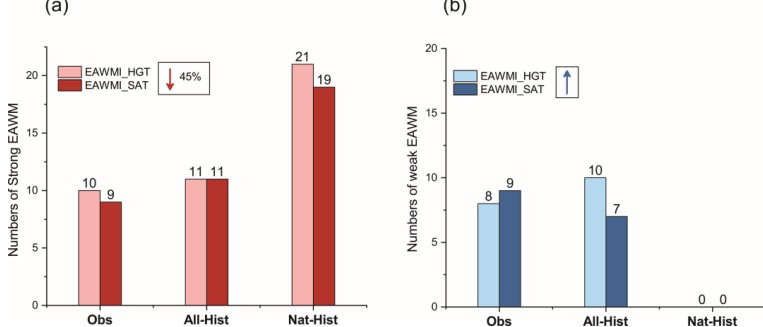

Figure 5 (a) The number of strong EAWM events during 1960–2012, based on NCEP
reanalysis dataset (left), outputs of model in the All-Hist runs (middle), and
outputs of model in the Nat-Hist runs (right). (b) As in (a), but for weak EAWM
events.