# Peer review of "Quantifying the contribution of anthropogenic influence to the"

_Atmospheric Chemistry and Physics, 2019_

## Referee Comment (RC1) · Anonymous Referee #1 · 15 Mar 2019

Based on model simulation, this study attributed the weakening of the East Asian winter monsoon (EAWM) over the past decades to explore the contributions of anthropogenic forcing including greenhouse gases, anthropogenic and natural aerosols, ozone, solar, and land cover. Overall, the topic is clear and the manuscript is well-organized and easy to follow. However, I have several major concerns before the manuscript is suitable for publication.

1. Numerous studies have shown that the EAWM had been gone through a significant weakening in the past few decades. CMIP5 model output was often used to identify that the weakness of EAWM was the response to global warming in the current and

future climate. Specifically, the change in the EAWM in future climate is considered to be a response to anthropogenic forcing. Compared to the previous studies, what is the new result from the current study?

2. This study is aimed to quantitatively estimate the contribution of anthropogenic forcing to the change in EAWM by one model output. It is hard to trust the results from a quantitative analysis of this type of study. Are the results robust or sensitive to models, especially the result shown in Fig. 5?

3. The introduction is not comprehensive and a number of relevant works have not been cited. On decadal time scale, the EAWM weakened in the late 1980s, but re-amplified after early 2000s (Wang and Chen 2014; Huang et al. 2014; Ding et al. 2014, 2015). For the causes of the decadal evolution of EAWM, many studies have shown that the changes in the Ural blocking and reduced Arctic sea ice are the main drivers (Wang and Chen 2014b; Mori et al. 2014; Luo et al. 2016).

Minor points:

1. The caption of Fig. 1d should be "Model-HGT". 2. Model's All-Hist runs can reproduce the climatology very well (Fig. 1), but fail to show the re-amplification of EAWM after early 2000s (Fig. 2a). It may lead to an overestimation of the contribution by anthropogenic forcing.

---

## Referee Comment (RC2) · Anonymous Referee #2 · 21 Mar 2019

The manuscript investigated the contribution of anthropogenic influence to the EAWM by the simulation in the All-Hist and Nat-Hist experiment. And they found that the weakening of EAWM in 1960-2012 is mainly attributed to the anthropogenic influence, especially in the frequency of strong EAWM event.

In general, I found the paper appropriate for ACP. However, it need to be minor revised before accepted this paper for publication in ACP with addressing those comments listed below:

Major Comments: 1. Although Nat-Hist and All-Hist experiment is well designed to investigate the impact of the natural forcing and anthropogenic forcing, I still suspect

[Figure]

that wether the SST in Nat-Hist overestimate or underestimate the influence of anthropogenic forcing and induce uncertainty. Therefore, I think the author should give discuss the uncertainty and explain why there is an increase trend in EAWMI in Nat-Hist. is it reasonable?

2. Line 186-187 and Fig. 5, "the case with the normalized index larger than 1.0 (smaller than -1.0) is defined as a strong(weak) EAWM event" but why the situation with zero weak event exist? I think may be due to "Note that the time series of the EAWM indices base on outputs of model in the Nat-Hist runs are standardized by the climatology simulated by the All-Hist runs." (the Line 435-436). This operation induce the averaged value of EAWMI in Nat-Hist is a positive value, so there is zero weak event. I think the author should explain why should standardized EAWMI in Nat-hist by the climatology of All-Hist. If standardized by itself climatology, does the conclusion of strong event decrease 45% also exist?

Minor Comments: 1. Line 420, should be "(b), (d) as in (a), (c)", and the title in Fig. 1d should be "Model-HGT"

2. Line 179-181, "an increase of SLP in the high-latitude East Asia" is contract with "the change of SLP also indicate a weak decrease of the Siberian high and an intensified Aleution low.", Based on the fig. 4c, the latter should be right.
* * *

---

## Author Response (AR1)

**Dear Editor:**

We would like to thank you for your effort on our manuscript. Based on the reviewers' comments, we have carefully revised the manuscript and marked the changes associated with replies in blue.
Sincerely,

Xin Hao

**Anonymous Referee #1**

Based on model simulation, this study attributed the weakening of the East Asian winter monsoon (EAWM) over the past decades to explore the contributions of anthropogenic forcing including greenhouse gases, anthropogenic and natural aerosols, ozone, solar, and land cover. Overall, the topic is clear and the manuscript is well-organized and easy to follow. However, I have several major concerns before the manuscript is suitable for publication.

**1. Numerous studies have shown that the EAWM had been gone through a significant weakening in the past few decades. CMIP5 model output was often used to identify that the weakness of EAWM was the response to global warming in the current and future climate. Specifically, the change in the EAWM in future climate is considered to be a response to anthropogenic forcing. Compared to the previous studies, what is the new result from the current study?**

Reply: Thank you for your comments. As you say, previous studies explored the response of the EAWM to global warming and revealed that the EAWM is weakened under different global warming scenarios. These researches are qualitative descriptions of the influence of the global warming scenarios on the EAWM. Their results did not provide the exact influence of the anthropogenic forcing on the EAWM in the past few decades. To investigate the causes of the weakening of the EAWM in the past decades, we quantitatively estimate the contribution of the anthropogenic emissions to the change of the EAWM in this study. In the All-Hist scenario, HadGEM3-A-N216 model was forced by historical anthropogenic and natural external forcing plus observed sea surface temperature and sea ice. In the Nat-Hist runs, anthropogenic forcings and land cover/use were set to preindustrial levels, and anthropogenic contributions to the observed SSTs and sea ice were removed. By comparing with two experiments, the results reveal the responses of the EAWM to anthropogenic forcings which are close to the observation values.

**2. This study is aimed to quantitatively estimate the contribution of**

**anthropogenic forcing to the change in EAWM by one model output. It is hard to trust the results from a quantitative analysis of this type of study. Are the results robust or sensitive to models, especially the result shown in Fig. 5?**

Reply: Thank you for your comments. In the supplementary material, we provide performance assessment of the EAWM in All-Hist runs. In the All-Hist scenario, HadGEM3-A-N216 model was forced by historical anthropogenic and natural external forcing plus observed sea surface temperature and sea ice. The results show that each All-Hist run with different initial state can reproduce the climatology very well and capture the increasing trend of the EAWM in the past decades reasonably. Most of the All-Hist runs can reproduce the decadal variability of the EAWM. Moreover, the ensemble-mean of the runs number 1, 2, 5, 13, 14 and 15 show a good performance in simulating interannual, decadal and linear trend change of the EAWM. It turns out the HadGEM3-A-N216 model can reliably reproduce the EAWM in All-Hist runs. Thus, we think the quantitative analysis in this study is reliable.

In this study, we define a threshold of 1.0 (-1.0) for the strong (weak) cases. Additionally, we also checked the results based on different thresholds (for example 0.8/-0.8 and 0.5/-0.5) and found that the conclusions are similar (Fig. x1 and x2).

[Figure]

Figure X1 Same as the Fig. 5 in paper, but based on the thresholds of 0.8/-0.8

[Figure]

Figure X2 Same as the Fig. 5 in paper, but based on the thresholds of 0.5/-0.5

**3. The introduction is not comprehensive and a number of relevant works have not been cited. On decadal time scale, the EAWM weakened in the late 1980s, but reamplified after early 2000s (Wang and Chen 2014; Huang et al. 2014; Ding et al. 2014, 2015). For the causes of the decadal evolution of EAWM, many studies have shown that the changes in the Ural blocking and reduced Arctic sea ice are the main drivers (Wang and Chen 2014b; Mori et al. 2014; Luo et al. 2016).**

Reply: Thank you for your comments. We have supplemented our introduction as follows:

The EAWM experienced remarkable transitions, with clear weakening since mid-1980s and re-amplification after mid-2000s (e.g., Yun et al., 2018; Wang and Chen 2014). The decadal oscillations in sea surface temperature (SST) are generally considered as the major source of the decadal variability of the EAWM, such as Pacific decadal oscillation and Atlantic multidecadal oscillation (Hao et al., 2017; Ding et al., 2014; Li and Bates, 2007). Jun and Lee (2004) suggested that the Arctic Oscillation may also contribute to the decadal variability in the EAWM. Additionally, above primary components of the EAWM system are subject to obvious changes under the influence of global warming (e.g., Li et al., 2018; Li et al., 2015; IPCC, 2013; Hori and Ueda, 2006; Kimoto, 2005; Zhang et al., 1997). Under different global warming scenarios, thermodynamic contrast between the East Asian continent and the Pacific Ocean is reduced uniformly characterized with weakening of the East Asian trough (EAT) as well as the East Asian jet, indicating a weakening of the EAWM (e.g.,

Xu et al., 2016; Kimoto, 2005). Previous studies based on Coupled models generally agree on the effect of global warming on the EAWM (Gong et al., 2018; Miao et al., 2018; Hong et al., 2017; Xu et al., 2016; Kimoto, 2005; Hu et al., 2000). Using the phase 5 of the Coupled Models Intercomparison Project output, Miao et al. (2018) deduced that both increased greenhouse gas concentrations and natural forcings (volcanic aerosols and solar variability) play key roles in the interdecadal weakening of the EAWM in the mid-1980s.

**Minor points:**

1. The caption of Fig. 1d should be "Model-HGT".

Reply: Thank you for your comments. We have revised the mistake.

2. Model's All-Hist runs can reproduce the climatology very well (Fig. 1), but fail to show the re-amplification of EAWM after early 2000s (Fig. 2a). It may lead to an overestimation of the contribution by anthropogenic forcing.

Reply: Thank you for your comments. As our results shown, HadGEM3-A-N216 can reproduce the climatology and decadal variability of the EAWM, including the re-amplification of EAWM after mid-2000s.

**Anonymous Referee #2**

The manuscript investigated the contribution of anthropogenic influence to the EAWM by the simulation in the All-Hist and Nat-Hist experiment. And they found that the weakening of EAWM in 1960-2012 is mainly attributed to the anthropogenic influence, especially in the frequency of strong EAWM event. In general, I found the paper appropriate for ACP. However, it needs to be minor revised before accepted this paper for publication in ACP with addressing those comments listed below:

**Major Comments:**

**1. Although Nat-Hist and All-Hist experiment is well designed to investigate the impact of the natural forcing and anthropogenic forcing, I still suspect that**

**weather the SST in Nat-Hist overestimate or underestimate the influence of anthropogenic forcing and induce uncertainty. Therefore, I think the author should give discuss the uncertainty and explain why there is an increase trend in EAWMI in Nat-Hist. is it reasonable?**

Reply: Thank you for your comments. We processed the difference of SST forcing between the All-Hist runs and Nat-Hist runs by empirical orthogonal function (EOF) analysis as EOF1 (Fig. 6a) and associated principal component 1 (PC1; Fig. 6b). The first leading mode shows a long-term oceanic warming with explained variance of 91.4%, characterized by negative anomalies in high-latitude oceans of the southern hemisphere, positive anomalies in tropical oceans and mid-latitude oceans of the southern hemisphere and intense positive anomalies in the high-latitude oceans around $60^o$ N. Figure 6c and 6d show the second leading mode of the observed SST obtained from Hadley Centre (downloaded from https://www.metoffice.gov.uk/hadobs/hadisst/; Rayner et al., 2003) by rotated EOF analysis. The second leading mode shows similar intensity and characteristics in the long-term oceanic warming with the response of SST to anthropogenic emissions. However, a cooling occurred in the northern Pacific and an obvious warming over Kuroshio region, which didn't capture by the models, may weaken the EAWM (Sun et al., 2016). Thus, this difference may induce an underestimation of the EAWM in Nat-Hist runs.

[Figure]

Figure 6 (in paper) The first leading mode (EOF1; a) and associated principal component (PC1; b) of the difference of the winter-mean sea surface temperature forcing between the All-Hist runs and Nat-Hist runs by empirical orthogonal function analysis based on the period of 1960-2013. The second leading mode (REOF2; c) and associated principal component (RCP2; d) of the winter-mean sea surface temperature from the HadISST data by rotated empirical orthogonal function analysis based on period of 1960-2013.

Previous studies indicated that the Atlantic multidecadal oscillation (AMO) and Pacific decadal oscillation favor a low-frequency variability of the EAWM, and that is the EAWM is weakened (enhanced) during the warm (cold) phase of the AMO/PDO (e.g., Li and Bates 2007; Ding et al., 2014; Hao and He, 2017). As shown in Fig. 2, an obviously increasing in EAWMI during 1960-1980 in Nat-Hist runs. During 1960-1980, both the PDO (downloaded from http://research.jisao.washington.edu/pdo/PDO.latest.txt) and AMO (Trenberth and Shea, 2006) were in a cold phase (Fig. S2), leading an enhanced EAWM. However, the PDO and AMO were out-of-phase after 1980s, causing a combined effect on the EAWM. Thus, we consider that the AMO and PDO may be responsible for the increase trend of EAWMI in Nat-Hist runs.

Related discussions have been supplemented in paper.

[Figure]

Figure S2 (in supplement) Time series of the Pacific decadal oscillation (PDO; a) and Atlantic multidecadal oscillation (AMO; b) during 1960-2012.

Reference:

Rayner, N. A.; Parker, D. E.; Horton, E. B.; Folland, C. K.; Alexander, L. V.; Rowell, D. P.; Kent, E. C.; Kaplan, A. (2003) Global analyses of sea surface temperature, sea ice, and night marine air temperature since the late nineteenth century J. Geophys. Res.Vol. 108, No. D14, 4407 10.1029/2002JD002670

Li, S. L., and G. T. Bates, 2007: Influence of the Atlantic multidecadal oscillation on the winter climate of East China. Adv. Atmos. Sci., 24, 126–135,

doi:10.1007/s00376-007-0126-6.

Hao, X, and S. P. He. (2017) Combined effect of ENSO-like and Atlantic multidecadal oscillation SSTAs on the interannual variability of the East Asian winter monsoon. Journal of Climate, 30, 2697–2716.

AMO Index Data provided by the Climate Analysis Section, NCAR, Boulder, USA, Trenberth and Shea (2006).

Sun, J. Q., Wu, S., and Aao J., Role of the North Pacific sea surface temperature in the East Asian winter monsoon decadal variability, Clim. Dyn., 46, 3793-3805, 2016.

**2. Line 186-187 and Fig. 5, "the case with the normalized index larger than 1.0 (smaller than -1.0) is defined as a strong (weak) EAWM event" but why the situation with zero weak event exist? I think may be due to "Note that the time series of the EAWM indices base on outputs of model in the Nat-Hist runs are standardized by the climatology simulated by the All-Hist runs." (the Line 435-436). This operation induce the averaged value of EAWMI in Nat-Hist is a positive value, so there is zero weak event. I think the author should explain why should be standardized EAWMI in Nat-Hist by the climatology of All-Hist. If standardized by itself climatology, does the conclusion of strong event decrease 45% also exist?**

Reply: Thank for your comments. The climatology of the EAWM in the All-Hist runs is very close to the results of reanalysis data, but larger than the climatology in the Nat-Hist runs. It would be more reasonable that the strong/weak events are defined on the same standard, so the EAWMI in the Nat-Hist runs are standardized by the climatology simulated by the All-Hist runs.

**Minor Comments:**

1. Line 420, should be "(b), (d) as in (a), (c)", and the title in Fig. 1d should be "Model-HGT"

2. Line 179-181, "an increase of SLP in the high-latitude East Asia" is contract with "the change of SLP also indicate a weak decrease of the Siberian high and an

intensified Aleution low.", Based on the fig. 4c, the latter should be right.

Reply: Thank you for your comments. We have revised the mistakes.

---

## Author Response (AR2)

This paper explores the contribution of anthropogenic influence to the EAWM in the past decades, using the All-Hist and Nat-Hist experiments. They found that the weakening of EAWM in 1960-2012 is mainly attributed to the anthropogenic influence, especially in the frequency of weak EAWM occurrence. Their results are reliable, based on the good performance of the model in simulating EAWM. I suggest for publication after minor revision. The details are shown below:

**1. As shown in Figure 2, the EAWM indices in the All-Hist runs during 1960-1970 disagree with the results from reanalysis data. However, the indices during 1970-2013 are closely related to that from reanalysis data. I think it may be due to the uncertainty of the NCEP dataset before 1970. To confirm the relationship, please check the performance of the EAWM indices in the All-Hist runs compared with JRA-55 reanalysis dataset.**

Reply: Thank for your comments. We have check the performance of the EAWM indices in the All-Hist runs compared with JRA-55 reanalysis dataset, and the results show similar characteristics (Figure R1 and Table R1).

**Table R1** "tr" is an abbreviation for "linear trend coefficient" (EAWMI_HGT/EAWMI_SAT). "cor" is an abbreviation for "correlation coefficient between simulated EAWM index under All-Hist scenario and observed EAWM index" (EAWMI_HGT/EAWMI_SAT), "cor_dec" is an abbreviation for "correlation coefficient in decadal time-scale". As a reference, the linear trend coefficient of EAWM_HGT/EAWM_SAT is -0.02/-0.023. The red numbers are significant at the 90% confidence level.

| | ensemble_best & JRA55 | ensemble_best & NCEP |
|---|---|---|
| Cor | 0.31/0.3 | 0.31/0.3 |
| Cor_dec | 0.73/0.69 | 0.76/0.7 |
| tr | -0.038/-0.044 | -0.038/-0.044 |

[Figure]

**Figure R1** Figure 2 (a) The time series of the normalized EAWMI_SAT (curve) and their linear trend (line) during 1960–2012, based on JRA55 reanalysis dataset (top), outputs of model in All-Hist run (middle), and outputs of models in Nat-Hist run (bottom). (b) As in (a), but for the EAWMI_HGT. "tr" is an abbreviation for "linear trend coefficient"."*"means the tr is significant at 95% confidence level based on the Mann-Kendall test, and "' " means the tr is significant at 90% confidence level. "cor" is an abbreviation for "correlation coefficient between simulated EAWM index under All-Hist scenario and observed EAWM index", "cor_dec" is an abbreviation for "correlation coefficient in decadal time-scale". Note that the time series of the EAWM indices base on outputs of model in the Nat-Hist runs are standardized by the climatology simulated by the All-Hist runs.

**2. Why the time series of the EAWM indices in the Nat-Hist runs are standardized by the climatology simulated in the All-Hist runs? Does it matter the number of the strong or weak EAWM events?**

Reply: Thank for your comments. The climatology of the EAWM in the All-Hist runs is very close to the results of reanalysis data, but larger than the climatology in the Nat-Hist runs. It would be more reasonable that the strong/weak events are defined on the same standard, so the EAWMI in the Nat-Hist runs are standardized by the climatology simulated by the All-Hist runs.

**3. According to previous studies (Zhu et al. 2015; Wei et al. 2017...), climate-decadal variability (such as PDO) associated with SST is important for the change of East Asian summer monsoon and winter monsoon. This paper indicates that the anthropogenic**

influence may be the main factor for the weakening of EAWM in 1960-2013, so what is the contribution of climate decadal variability related to SST? Is it smaller than the anthropogenic influence?

Reference:

Zhu Y, Wang H, Ma J, Wang T, Sun J. 2015. Contribution of the phase transition of Pacific Decadal Oscillation to the late 1990s' shift in east china summer rainfall. J. Geophys. Res. 120:8817–8827.

Wei Y, Yu H, Huang J, He Y, Yang B, Guan X, Liu X (2017) Comparison of the Pacifc Decadal Oscillation in climate model simulations and observations. Int J Climatol. https://doi.org/10.1002/joc.5355

Reply: Thank for your comments. There is no doubt the PDO is an important reason for the decadal variation of the EAWM. As shown in Fig. 2, an obviously increasing in EAWMI during 1960-1980 in Nat-Hist runs. During 1960-1980, both the PDO and AMO were in a cold phase (Fig. S2), leading an enhanced EAWM. However, the PDO and AMO were out-of-phase after 1980s, causing a combined effect on the EAWM. Thus, we consider that the AMO and PDO may be responsible for the increase trend of EAWMI in Nat-Hist runs. In All-Hist runs, there is an obvious weakening of the EAWM during 1960-2013. In this paper, we think the anthropogenic influence is the essential factor for the linear trend (a weakening) of the EAWM in 1960-2013.

**4. Line 17, "… monsoon can be greatly influenced …" can be changed to "… monsoon is greatly influenced …".**

Reply: Thank you for your comments. We have revised the mistakes.

**5. Lines 186-188, "Meanwhile, in the high-level troposphere, … over the high-latitude regions under the anthropogenic influence"; Line 195, " a decrease of SLP in the mid-latitude East Asia". I suggest that more details should be provided in these descriptions.**

Reply: Thank you for your comments. We have revised the mistakes.

**6. Line 204, "Interestingly, the two simulated EAWM indices …". "Interestingly" is redundant.**

Reply: Thank you for your comments. We have revised the mistakes.

**7. Line 238, "the interannual and interdecadal variation of the EAWMI_HGT…". The**

**"variation" should be "variations".**

Reply: Thank you for your comments. We have revised the mistakes.

**8. Line 489, "… and 850 wind …" should be "… and 850 hPa wind …".**

Reply: Thank you for your comments. We have revised the mistakes.